# Children, caregivers and health workers' perceptions and experiences of the XTEMP-R tool to improve tuberculosis treatment

Dillon T. Wademan[1]*, Willdon J. Filander[1], Mfundo Mlomzale[1], Ntokozo Sibisi[2], Cyril Thwala[2], Phumlani Memela[2], Nosivuyile Vanqa[1], Megan Palmer[1], Tina Sachs[1], Munira Khan[2], Rajneesh Taneja[3], Poonam Pande[3], Koteswara Rao Inabathina[3], Anneke C. Hesseling[1], Anthony J. Garcia-Prats[1,4], Graeme Hoddinott[1,5]

1 Desmond Tutu TB Centre, Department of Paediatrics and Child Health, Faculty of Medicine and Health Sciences, Stellenbosch University, Cape Town, South Africa, 2 Tuberculosis and HIV Investigative Network (THINK), Durban, South Africa, 3 Global Alliance for TB Drug Development, New York, New York, united States of America, 4 Department of Paediatrics, University of Wisconsin School of Medicine and Public Health, Madison, Wisconsin, United States of America, 5 Sydney School of Public Health, Faculty of Medicine and Health, University of Sydney, Camperdown, Australia

* dtwademan@sun.ac.za

## Abstract

Treating drug-resistant tuberculosis (DR-TB) in children remains a significant challenge for patients, caregivers, and health systems, despite advances in child-friendly drug formulations. While new formulations offer benefits, their widespread availability is limited, and many exhibit poor palatability. A key strategy to improve administration and mask the taste of paediatric TB medications involves creating extemporaneous suspensions. However, this often requires pharmaceutical services not readily available in high-burden settings. To address this, the Global Alliance for TB Drug Development (TB Alliance) developed XTEMP-R, an inexpensive prototype tool designed to facilitate home-based preparation of liquid TB medication suspensions. This study explored the experiences and perceptions of children, their caregivers, and health workers regarding the XTEMP-R tool for preparing extemporaneous DR-TB treatment suspensions. We collected qualitative data from two sites in South Africa. The first component involved interviews with 17 caregivers and 12 health workers, followed by focus group discussions, with participants directly interacting with the XTEMP-R tool. The second component comprised 31 interviews with 11 caregivers of 13 children who used the XTEMP-R tool for home administration. Case descriptions were iteratively refined and analyzed using deductive thematic analysis. Findings indicate that children, caregivers, and health workers found the XTEMP-R tool easy to use, clean, and store, appreciating its appealing color and durability. Home users reported that the tool simplified treatment preparation and administration, reducing time and relational burdens associated with DR-TB treatment. While XTEMP-R effectively addressed usability challenges related to drug preparation, fundamental

**Data availability statement:** The datasets generated and analysed during the current study are not publicly available due to the need to protect participant confidentiality. The data contain sensitive and potentially identifying patient information. Data are therefore available upon reasonable request. Requests can be directed to the Health Research Ethics Committee at Stellenbosch University (ethics@sun.ac.za).

**Funding:** This study was made possible through Unitaid's funding of the BENEFIT Kids project to Stellenbosch University. Unitaid accelerates access to innovative health products and lays the foundations for their scale-up by countries and partners. GH is supported by funding from financial assistance of the European Union (Grant no. DCIPANAF/2020/420- 028), through the African Research Initiative for Scientific Excellence (ARISE), pilot 475 programme. ARISE is implemented by the African Academy of Sciences with support from the European Commission and the African Union Commission. The contents of this document are the sole responsibility of the author(s) and can under no circumstances be regarded as reflecting the position of the European Union, the African Academy of Sciences and the African Union Commission. The funders had no role in study design, data collection and analysis, decision to publish, or preparation of the manuscript.

**Competing interests:** The authors have declared that no competing interests exist.

obstacles concerning medication palatability, nausea, and side effects remain significant barriers. Importantly, the tool appeared to foster increased treatment responsibility among some children, suggesting a potential pathway to improve therapeutic engagement and agency. This research underscores the XTEMP-R tool's potential to ease paediatric DR-TB treatment and highlights crucial areas for design refinement, ultimately aiming to enhance adherence and overall outcomes.

## Introduction

Over 1 million children develop tuberculosis (TB) each year [1]. The burden of drug-resistant (*Mycobacterium tuberculosis* resistant to at least one anti-tuberculosis drug) (DR-) TB is not well described in children and adolescents; model-based estimates suggest that at least 25,000–32,000 children < 15 years of age develop multi drug-resistant tuberculosis, *Mycobacterium tuberculosis* resistant to at least isoniazid and rifampicin, (MDR-TB) each year globally [2]. DR-TB treatment for children remains complex, despite recent advances in drug regimens, and is a priority for children affected by TB, their caregivers and health care providers to ensure access to better, safer, and more child-friendly treatment for DR-TB [3–5]. As the diagnosis and treatment of *Mycobacterium tuberculosis* resistant to rifampicin (RR-)/MDR-TB are scaled up globally, an increasing number of children stand to benefit from better treatments.

Despite the recent development of more 'child-friendly' formulations of some second-line TB medications such as levofloxacin, moxifloxacin, clofazimine, linezolid, bedaquiline and delamanid, many of these 'child-friendly' formulations are not yet widely available and some remain poorly palatable [6]. Even when these formulations are available, the processes involved in preparing, administering, and adhering to treatment for children who struggle to swallow tablets or where dose titration is needed, remains problematic [7]. Children must typically ingest upwards of 6 tablets every day for months [4].

In many resource-limited or household settings, pre-manufactured liquid suspensions and syrups are not readily accessible or easily prepared. One strategy to overcome barriers to administration and improve the pragmatic taste masking for TB medicines among children, in the field is by creating extemporaneous suspensions, where a tablet or powder is mixed into a liquid suspending agent and the suspension is administered to the child orally with a spoon, syringe, or a dispenser. Extemporaneous suspensions can comprise of multiple medications blended together into a homogenous solution. Such suspensions are often the preferred formulation type for young children ≤10-years-old and are widely used in settings with high burden of TB [8]. In these settings, pharmaceutical services needed to prepare extemporaneous formulations are typically limited. For pharmaceutical and health systems purposes, solid oral dosage formulations of medicines are preferred as they are easier to transport, store, and have a longer shelf-life [9]. Extemporaneous solutions may also pose challenges with regards to homogeneity, bioavailability and dosing accuracy [10].

Recently, the Global Alliance for TB Drug Development (TB Alliance) developed an inexpensive prototype tool called 'XTEMP-R', to facilitate the creation of home-based

homogenous liquid suspensions of TB medications, where treatment administration usually occurs, thus avoiding the need for pharmacy services [11]. The XTEMP-R is a cylindrical shaped object with a flexible receptacle, tight-fitting cap, and a suction cup bottom (Fig 1). It is a soft walled tool manufactured with biocompatible, low hardness high-consistency silicone rubber. The height of the XTEMP-R tool is 140 mm, the internal diameter is 22 mm, and the capacity is approximately 40 mL. The tool has volume markings on the outside at 5-mL intervals starting at 15 mL. This tool is resistant to extremes of environments and temperatures and is designed to be leak-resistant and is easy to squeeze between the fingers for manual dispersion of the tablets.

Extemporaneous suspensions of bedaquiline, clofazimine, pretomanid and delamanid have been successfully tested for stability, offering a viable alternative for people who have difficulty swallowing including children and adults [12–15]. We previously tested a suspension of these four drugs using the XTEMP-R tool and found it feasible and safe to accurately dose children with DR-TB [11]. The use of stable suspensions of TB regimens may also improve the overall acceptability TB treatment for children and their caregivers. Improved treatment experiences are valuable in themselves, but may also have the instrumental benefits of improved adherence, leading to better treatment outcomes, and lower overall health systems costs [16,17]. More acceptable TB regimens align with the first pillar in the World Health Organisation's (WHOs) End TB Strategy, which is to achieve integrated, patient-centred care [18].

In this study, we sought to understand children, their caregivers and health workers' experiences of using the XTEMP-R tool for the preparation of extemporaneous suspension formulations to administer DR-TB treatment in the household (S1 Fig, S1 File). We (1) describe caregivers' and health workers' perceptions of the XTEMP-R tool, (2) describe how well the tool integrated into children and caregivers' daily routines for preparing and administering medication for antituberculosis treatment, and (3) reflect feedback from children and caregivers on how to improve the XTEMP-R prototype for future use

## Methods

### Ethics Statement

Stellenbosch University's Health Ethics Research Committee provided ethical approval for the study (N23/07/080). Children 0–6-years-old were asked verbally if they were willing to interact with the study team, along with caregivers written

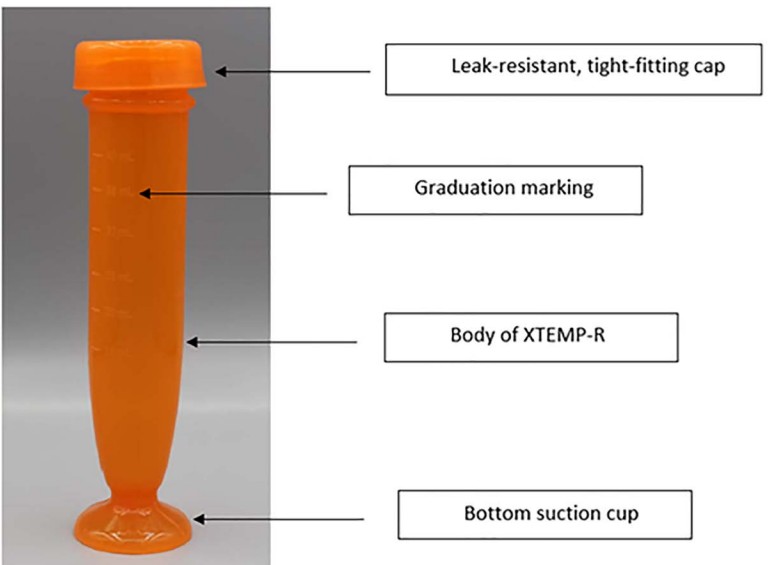

**Fig 1. Image of the XTEMP-R tool.**

consent. Children 7–12-years-old completed written assent along with caregiver consent. Healthcare workers provided written informed consent prior to participation, after permission to approach them was obtained from site and healthcare facility managers.

## Setting

The study was conducted in the Cape Metropolitan area of the Western Cape province and in the uMgungundlovu and eThekwini districts in KwaZulu-Natal province South Africa. Both settings have a high burden of TB and DR-TB. Together, these areas are home to ~10 million residents, living in rural to peri-/urban communities, and report some of the highest TB burdens in the world. Most households affected by TB live in high-density, low income, peri-urban townships comprising of mostly informal housing with relatively poor access to municipal services and considerable population mobility [19–21]. The two clinical research sites, in Cape Town, Western Cape Province and in Durban, KwaZulu-Natal, have teams with extensive experience in clinical research among children with DR-TB. However, the pathways to care differ significantly between these provinces. In KwaZulu-Natal, children are typically referred to a centralized hospital for the entire duration of their treatment. Conversely, ambulatory decentralized care is the standard of care for children with DR-TB in the Western Cape. In both provinces, children are hospitalized—or remain so—if their care is compromised by psychosocial, economic, or clinical complexities. Children recruited through these sites received the nationally recommended RR/MDR-TB treatment regimen at each site.

## Data collection

The study included two components. In the first component, we interviewed health workers and children's caregivers between February and June 2024. We conducted in-depth interviews and a focus group discussion with these participants. In this component we showed caregivers and health workers how the XTEMP-R tool works and discussed the feasibility of its use in the context of the household. We recruited health workers with at least one year of experience in providing care to children affected by DR-TB, ranging from research counsellors/community health workers to medical specialists. We recruited caregivers who had experience of preparing and administering DR-TB treatment to children during the past year. We used maximal variation sampling to purposively recruit caregivers whose children had a diverse range of DR-TB experiences across kay variables, including their child's age, sex, and language (as a proxy for ethnicity). In this way, we sought a heterogenous sample to improve the transferability of our findings [22]. Findings from Component One informed our initial interactions with caregivers and children in Component Two, especially regarding managing expectations and providing clear instructions for the XTEMP-R tool.

The second component involved a qualitative, longitudinal cohort design. Caregivers of children on DR-TB treatment were recruited and ~3 interactions per child-caregiver dyad were conducted over 4–8 weeks, between May-November 2024. Through Component two we sought to (a) understand how caregivers and their children used the XTEMP-R in their everyday treatment routine, and (b) inform refinement of the XTEMP-R tool prototype, including how its use is explained to future potential users. Children on DR-TB treatment aged 0–12-years-old who were not routinely able to swallow or administer their tablets whole, along with their caregivers, were eligible for participation. We gave caregivers the XTEMP-R tool to use to support administration of their treatment at home. As in component one we used maximal variation sampling to purposively sample caregiver and child participants to maximise diversity in child age, sex and language. Each interaction with children and their caregivers involved an in-depth interview using a semi-structured discussion guide that included participatory activities encouraging participants' active engagement and illustrating practical operation of the XTEMP-R tool in the household setting [23]. Interaction topic areas included (a) how caregivers and children had incorporated the use of the tool into their treatment administration routine, (b) challenges caregivers/children experienced using the XTEMP-R tool, and (c) recommendations caregivers/children had for potential future XTEMP-R tool users. Socio-behavioural researchers (DTW, MM, WJF, NS, CT, PM) conducted the interviews locally in participants' households

(or other convenient/ favoured spaces) in participants' preferred language (Afrikaans, English, *isiXhosa*, or *isiZulu*). Each recorded interaction lasted between 30–60 minutes. Researchers completed detailed case descriptions immediately after each interaction with participants, following structured documents.

## Data analysis

Data collection and analysis was iterative and ongoing, and involved comparing each interview and emergent codes with previously collected data. Data collection and analysis continued until we reached code saturation – the point at which new data collection was not yield new insights [24]. Case descriptions facilitated consideration of complexity and context-specificity of participants' experiences, and were informed by *in situ* field notes of participatory observations, supplemented with verbatim quotes collated from audio recorded discussions [25]. Only once each case description had been reviewed by all researchers was it considered complete. A single case file document compiled completed case descriptions from each participant. Case files formed the primary dataset towards developing a codebook using an acceptability framework to guide deductive thematic analysis [26].

## Findings

For component one, we conducted individual interviews with 17 caregivers and 12 health workers, and two focus group discussions each with caregivers and health workers. For component two, we conducted 31 interviews with 11 caregivers of 13 children on treatment. Nine children were girls, five spoke Afrikaans, six Xhosa and two Zulu. The median age of children was 4.5 years-old (IQR 2.25 – 6.5).

## Component 1: Caregivers' and health workers' perceptions of the XTEMP-R tool

At first, both caregivers and health workers were confused by the XTEMP-R tool, saying they were uncertain what it is and how it is used. However, after the research team had explained how the tool is used, caregivers and health workers quickly saw its potential. For example, one health worker said,

> "So instead of using an old-fashioned pestle and mortar, or crushing it on a saucer with a spoon? Yeah, that would be ideal, because I've been there: crushing [pills], and it's fine. But invariably you get spillage […] It's never fully accurate, and then you often have to transfer [the powder] to something else, whereas this [the tool] is ideal" (Clinician).

Healthcare workers believed the tool would improve the preparation and administration process involved in DR-TB treatment, especially for caregivers of young children. However, one health worker saw the potential of its use by older children,

> "I think bigger [older] kids would be able to use it by themselves, but then I'm not talking about 8-year-olds, 9-year-olds, I'm talking about a little bit bigger [older] but then the other thing is […] they're swallowing already. […] you will have that some children [such as] the bigger [older] ones [who] don't want to swallow tablets" (Nurse)

Health workers and caregivers especially liked the tool's silicone composition, that it is malleable, has a suction cup bottom and is not prone to breaking or spilling the treatment.

"It's easy because it's safe. Because even if you leave it to dissolve it's closed, nothing can get into it. It won't spill and dust won't get in because […] it has its own lid" (Caregiver, 5Y,M).

Many caregivers remarked how the tools' composition facilitated crushing children's treatment, making it easier to prepare and ingest. The caregiver of 7-year-old girl also remarked on the tool's potential among other people who may struggle to take treatment:

"It [the XTEMP-R tool] will help [children] a lot […] even old people don't finish treatment, because they see the pills and have to swallow them. You see when it's done like this, they can just drink it. It's like they're drinking water. […] It's nice that you can crush pills inside. Now it's like you're drinking water. It is right, it is very good [and] because it's rubber, it will last a long time" (Caregiver, 7Y, F).

Another caregiver of a three-year-old girl said that while the tool would be helpful to older children, she would still rely on a syringe to administer the treatment to her young daughter:

"I will maybe use a syringe, it's easier. […] Maybe five-year-olds can drink [the treatment]. But a syringe must be used for [children] younger than that age, because they will just spit it [the treatment] out" (Caregiver, 3Y,F).

The primary advantage of the tool, welcomed by caregivers, is that it facilitates 'crushing' pills. Although, only just introduced to the XTEMP-R tool, a caregiver immediately recognised its usefulness, explaining,

"Throw the [pills] in this thing and crush it […] add a bit of water and shake it […] [If] she crushes it like this; she wouldn't need a spoon because it [the XTEMP-R tool] is flexible to crush the medication and then you can administer it as is" (CHEXTC1Caregiver, 5Y,F).

However, caregivers and health workers were concerned that the volume markings started at 15 ml which they argued was too much for young children who routinely have their treatment administered in as little as 5–10 ml. One caregiver acknowledged that it would depend on how much liquid was prescribed for administering the treatment,

"It depends on the pills. I think that [volume] is enough because, people don't use too much water for one pill […] sometimes they give you instructions on how much water the child must drink with the pill" (Caregiver, 7Y,F).

Additionally, caregivers and health workers found the volume markings difficult to read and suggested using contrasting colours for the markings to improve their visibility,

"I can't see them now […] they are visible but not clearly visible. […] The words should be written in white so that they can be seen, not the same as the container, for example here in orange put another colour" (Caregiver, 5Y,M).

Caregivers and health workers reported that the tool would be easy to store, to clean and maintain. A few caregivers noted that they would keep the tool and use it in the future for other treatments for their children.
"I would keep it and use it in future for other treatments because it is very helpful, and everyone can benefit from the tool because older people also struggle with swallowing tablets" (Caregiver, 4Y,F).
Caregivers and health workers expressed varied reactions to the tool's colour. Some said that the bright orange colour would be appealing to children and would also disguise children's treatment's colour. However, others pointed out that the colour might also mask how much of the treatment had dissolved or been administered. As one father explained,

"it's too dark because sometimes you want to administer the treatment late in the evenings and people like me who wear glasses won't see it, or there is loadshedding. It would be better if it was transparent" (Caregiver, 5Y,M).

Caregivers and health workers were not concerned about the tool's durability but noted that its composition could potentially be damaged by cleaning it in hot water.

## Component 2: Overcoming Treatment Challenges with XTEMP-R

Overall, participants' initial reaction to the XTEMP-R tool was positive. As health workers and caregivers in Component 1 expressed – caregivers reported that the tool facilitated easier preparation, and administration to all children, but was especially useful to support those who struggle to swallow tablets. The caregiver of a two-year-old girl said, "Wow, it's nice to use, even when you are holding it, the way it's made, it's like she's holding a toy".

Another caregiver of three children concurrently on DR-TB treatment said that her middle-child, struggled to ingest the treatment more than her younger sister,

> "[My middle child] was the one who was giving me problems in the early stages of the treatment. She was vomiting after ingesting the tablets. Gosh! She was vomiting […] the others had no problem. […] The other siblings would even encourage her to take her medication, and she would drink them one by one. […] At first, I didn't know that you can crush the pills and [my middle child] did not want to drink the yellow tablet. However, when the yellow tablet was crushed and mixed with other tablets, she does not have any problems." (Caregiver, 18Y,M,8Y/5Y,F)

Before receiving the XTEMP-R tool, caregivers described the treatment preparation and administration process as being tedious. Caregivers consistently described having to crush the pills before administering their child's treatment. This was described as both a difficult and lengthy process, and concerns around losing treatment on the floor, inaccurate dosing, and poor hygiene were raised.

Crushing tablets with hands is not safe because they have germs. This tool can be easily used because it is very soft, has a lid, and the dust cannot get inside (Caregiver 2Y,F).

Another caregiver of a 5-month-old girl and a 4-year-old boy described a similarly difficult treatment preparation process she had devised before receiving the XTEMP-R tool. The caregiver described the process as a hygienic and simple method for crushing her children's medication. However, she reported that this process was time-consuming, taking approximately half an hour to prepare and administer treatment to both children. She explained the process as follows:

> "I prefer my method of crushing the pills because it is quicker. So, I put the pills in here [a plastic bag] and wrap a piece of paper around it and crush it [with a spoon]. Then it is finely crushed. […] This way I don't lose/spill any of the treatment".

Once she was done crushing her children's treatment into a 'fine powder,' she put it in a bit of yoghurt that she fed to her daughter. Her son, however, sometimes chewed his pills. The caregiver said that some pills made her children nauseous, and they often vomited them. Since receiving the XTEMP-R tool, however, the way she had prepared and administered treatment had changed. She explained that the reduced volume of liquid facilitated by mixing all the tablets together meant her children no longer vomited the medication:

> "I crush everything [pills] at once, I don't administer it one after each other. I add everything together in the instrument. […] I add the pills to the instrument and then I press it the whole time. I crush it like that. […] It helped. It helped me because when I use the spoons then everything came out [vomit] again. But when I use that [instrument] then nothing comes out [they do not vomit]" (Caregiver, 4Y/5Mon, M/F).

A major challenge highlighted by caregivers that the XTEMP-R tool helped them to overcome, was their inability to easily travel with their children's treatment or even administer their child's treatment while travelling.

"When you have a cup in the house […] it sits in the medicine cabinet […]. It is not easy to take it with you anywhere you go" (Caregiver, 2Y,F).

This was particularly important for caregivers living in rural areas who often travelled long distances to cities or even to the health facility. The caregiver of three children aged, two, four and eight-years-old, admitted that her youngest child would miss their medication doses when they were travelling because it was impossible to prepare and administer the treatment:

> "Once a month, I wake up very early in the morning to fetch my treatment and at that time she is still asleep, and I don't want to wake her up to give her the tablets. So, I just give her in the afternoon when I come back from the clinic" (Caregiver, 2Y,4Y,8Y,F).

Since being given the tool, however, the caregiver said she's able to administer the treatment whilst travelling,

> "Ever since I received the tool in October, and I carry it everywhere. Even in the taxi, I take it out and give her the tablets and when we are visiting the hospital, it is there with me" (Caregiver, 2Y,4Y,8Y,F).

Caregivers described how preparing and administering their child's treatment was often emotionally taxing, and that they had to come up with ways to overcome their children's reticence towards their treatment, often relying on encouragement, bribery, coercion or threat. One caregiver said,

We had to persuade him in certain ways, put some nice things here, let him drink, then eat something nice, you have to encourage him with something nice, something he can eat (Caregiver, 3Y,M).

Conversely, caregivers reported that because their children liked the tool so much, the entire treatment process had become easier. The caregiver of a three-year-old boy described how the tool had changed the way her son interacted with his treatment,

> "He has a new thing now that he doesn't want to drink from a syringe or a cup, he would say, pour on my thing […] that thing that is like a glass, it is mine" (Caregiver, 3Y,M)

Although caregivers attributed their child's interest in the tool, to its colour, malleability or 'playfulness,' with one caregiver saying her two-year-old son "is more impressed with the instrument than the colour" (Caregiver, 3Y,F) and another euphemistically, calling the tool her daughters' 'carrot' (Caregiver, 1Y,F). Other caregivers said that their child's interest in the tool went beyond this, explaining their children exerted greater ownership over their treatment process, ensuring they received treatment on their own terms. The caregiver of a two-year-old girl said,

> "She likes [the tool]! Even when there are visitors, it's more pleasant than when someone arrives holding a syringe. […] When [visitors] arrive, she says, 'Here's my pill,' and she even wants to play with [the tool], showing [the tool] to the other children, saying, 'these are my pills'" (Caregiver, 2Y,F)

The tool helped caregivers overcome some of the pragmatic challenges of DR-TB treatment preparation and administration. Perhaps more importantly, the tool's 'playful' characteristics facilitated children's involvement in their own treatment adherence, extending their level of agency of their care.

## Component 2: Limitations: Unresolved Challenges in DR-TB Treatment

Almost all the caregivers mentioned that their children complained about the palatability of their treatment. Although this was largely unchanged by the XTEMP-R tool, the caregiver of 2-year-old and 8-month-old girls described her strategy to identify which drugs were 'most bitter,' and how she adapted her administration process to minimise her children's distress,

 

At first, I thought let me give it [bitter pill] to her first because I thought it would taste nice. However, afterwards she told me, 'Mommy, it is bitter'. Then I gave her the other pills and she [also] told me 'No, mommy. It is bitter'. Then I thought I am just going to give all the pills together because I don't know which pill is bitter or how the other pills taste".

Non-dispersible tablets made treatment administration particularly difficult for caregivers, despite having the XTEMP-R tool. Caregivers frequently resorted to crushing, breaking, or cutting their children's pills at home and devising alternative administration techniques due to the scarcity of dispersible drug formulations. The caregiver of a three-year-old girl spoke about preparing and administering clofazimine in its gel-capsule form,

> "I added it to a cup and tried to prepare it, but it was almost like oil on water. I struggled to remove the residue. The next day I told her father, 'I am going to cut the head and see what is inside'. When I cut the head, I saw that I could have done it like that a long time ago: cut the head and squeeze [the contents] into her mouth. Then she swallowed it just like that. I just told her it is chocolate and that one smelled nice" (Caregiver, 3Y,F).

Although having access to dispersible tablets made it easier to prepare and administer their children's treatment, some of the dispersible drugs remained challenging to administer even when they could disperse treatment in the XTEMP-R tool, especially those requiring titrated dosing. For example, a two-year-old girl's mother went on to say,

> "At first, I had to crush [the pills] and each had varying amounts of water. Every pill had its own amount of water. When they changed it now, they made it easier for me, because there were times when I crushed the pill and it flew everywhere, but that [dispersible] pill you just add water and then it dissolves. But they also told me I can change the water, and I can mix the rest of her pills together and mix it with 7 to 8 mils of water. Those are four [of her] pills. She receives the Clofazimine on its own. And then it is the bedaquiline which she gets on Monday, Wednesday and Friday. [Bedaquiline] I also have to prepare on its own because it requires 10 ml of water and then she has to get 6 ml. [...]. I know that those [four] pills that she has to drink afterwards are bitter. Then I have to give it to her bit by bit. Because then she [won't] spit out the pills because I'm giving it all at once."

A further complication for very small children was that while the tool facilitated the rapid and hygienic dispersion of medication, syringes remained a necessity for those children too young or unable to drink their medication directly from the tool. The tool was given to caregivers along with a syringe adapter, which facilitated withdrawing the treatment from the tool, as described by the caregiver of two-year-old and nine-year-old girls said,

"I take 10ml water measure with a syringe then put it inside the tool. I then add my tablets, close the tool, press and shake whilst I wait for the tablet to dissolve [...] It is easy to withdraw from the tool as long as you have a syringe".

For caregivers of children old enough to use the tool themselves with dispersible treatments, the tool simplified preparation. However, it offered little help to caregivers of very young children using non-dispersible treatments. For these caregivers, treatment preparation often entailed crushing the pills outside of the tool, or individually preparing the drugs, like with the Clofazimine capsules.

## Components 1 & 2: Caregivers' recommendations for future use of the XTEMP-R tool

Caregivers expected future users to find the tool user-friendly and encouraged them to follow instructions on how to prepare and administer their children's treatment.

> "The first time my nieces came to visit, they asked me 'what is it?' [...] I told them it was the new instrument for [my son's] pills. They said, 'it is very nice'. [...] Then they told me they really like the instrument. [...] it is easy for me [to use] so I think it will also be easy for the next person if I perhaps show them how to use it" (Caregiver, 5Y,M).

Caregivers' and health workers' instructions on how to use the tool generally aligned with the developer's recommendations (S1 Fig). A few noteworthy exceptions were that caregivers recommended using a syringe to mete out the exact volume of water required to prepare the treatment, waiting 10–15 minutes for the tablets to soften before beginning to crush the tablets in the tool, and vigorously shaking the tool to speed-up the dispersion of tablets. A few caregivers made recommendations to make the tool and its use more child-friendly. For example, the caregiver of a 9-month-old girl recommended using breastmilk as the suspension vehicle and suggested bending the tool so that it would be easier to control how much of the suspension comes out at a given time. This control measure allowed the caregiver to use the tool instead of a syringe while administering her child's treatment:

> "I add the milk just like that [...] I don't pull it [suspension] up [with a syringe]. I administer it as is with that part [points toward the opening of the instrument] and then I bend that part." (Caregiver, 9Mon,F)

The consensus among caregivers and health workers, appeared to be that caregivers adapt their treatment preparation and administration processes to their children's needs. Many caregivers planned to keep the tool for future use in case any children in their care become sick. One caregiver went on to describe how she had used the tool to help administer treatment to her child who had recently suffered from diarrhoea.

> "I will keep it for her [Stacy], I wouldn't give it away. Because one can need it again. Like now with the water, I can use it again. And her stomach is still like that [runny]" (Caregiver, 3Y,F).

Other caregivers spoke about using the tool for other diseases as well, and for children of different ages. Overall caregivers and health workers believed that the XTEMP-R tool had great utility that could be easily integrated into everyday care practices and tailored for use in each household.

## Discussion

The XTEMP-R tool was taken up with enthusiasm by most caregivers of children who had received treatment for RR/MDR-TB. Healthcare workers and caregivers with prior experience of children on DR-TB treatment also liked the tool. The tool was described as appealing to children, appropriate in size and shape, of durable material, making it easy to clean and store. The tool helped caregivers overcome some of the pragmatic challenges common to DR-TB treatment in children irrespective of age. Specifically, it facilitated easier preparation and administration of children's treatment. However, caregivers and health workers reported that the tool could not overcome all challenges posed by caring for children on DR-TB treatment. On the positive side, some caregivers also reported a change in their children's treatment engagement, suggesting that after their children received the tool, they took more ownership of their treatment and care.

Overall, the XTEMP-R tool primarily ameliorated challenges caregivers and children experienced in the 'usability' domain of an acceptability framework [26]. Specifically, the tool eased preparation and administration of drugs for children and improved the appeal of treatment. Making treatment more appealing, including the drug or suspension colour, was an important finding in another qualitative study that explored children's preferences for MDR-TB [27]. The tool also helped caregivers overcome logistical issues related to treatment including when and where children ingested treatment. However, the tool did not alter the palatability of the drugs – this remains a significant challenge for caregivers and their children on DR-TB treatment [27]. Furthermore, the use of non-dispersible tablets remained challenging for caregivers and their children. The limited widescale access to dispersible drugs remains a major hindrance to better RR/ MDR-TB treatment for children [6]. Dispersible medications are consistently reported as being child-friendlier and preferable than non-dispersible treatment in children [28,29]. For example, as we have previously described, clofazimine capsules were repeatedly mentioned by caregivers and health workers as problematic DR-TB treatment [30].

The XTEMP-R tool may also reduce intra- and interpersonal challenges that often come with TB treatment [26]. Caregivers' reporting children's increased participation in their treatment process was an unexpected positive finding. Successfully including children in their own treatment and care is a significant challenge that cuts across the life-span of a chronic disease [31]. This is often limited by children's ability to understand their disease and treatment, as well as caregivers' ability to communicate this information [32]. Additionally, caregivers' own health literacy and perceptions of illness and disease can influence what and how they communicate with their children [33]. Caregivers and children, alike, often experience fear, anxiety and depression over the course of TB treatment, negatively impacting their combined willingness to remain in care [34].

Strengths of this study include its multi-stage approach and inclusion of two sites with diverse geographic and cultural profile, and the inclusion of data from health workers and caregivers, providing insight into the supply- and demand-side of TB treatment experiences among children. The longitudinal nature of study component 2 allowed caregivers sufficient time to appropriately integrate use of the XTEMP-R tool into children's everyday treatment regime. Although all health workers interviewed possessed substantial experience of care for children affected by DR-TB, their insights were often derived from clinical research care settings. Consequently, this study may not fully capture the challenges and barriers that health providers encounter within routine clinical practice in local TB programmes. In addition, our purposive recruitment strategy may have inadvertently favoured caregivers who were already more engaged with healthcare services, as they were involved in prior or ongoing studies. Therefore, this work may not fully detail the challenges experienced by caregivers in the wider population, particularly those who face significant barriers to accessing care.

In conclusion, this study highlights the persistent complexities of DR-TB treatment in children despite access to dispersible formulations. While the XTEMP-R tool effectively addressed usability challenges related to drug preparation and administration, fundamental obstacles concerning medication palatability and associated nausea remain significant barriers, particularly for younger children. Notably, the tool appeared to foster increased treatment responsibility among some children, suggesting a potential pathway to improve therapeutic engagement. Future research should investigate how enhancing children's agency and active participation throughout the treatment journey impacts their experiences, adherence, and overall outcomes, alongside those of their caregivers. Addressing these behavioural dimensions, in parallel with efforts to develop more child-friendly formulations, is crucial to improve TB care in children.

## Supporting information

**S1 Fig. Description of how to use the XTEMP-R tool.**
(DOCX)

**S1 File. Link to video demonstration of how to use the XTEMP-R tool.**
(DOCX)

## Acknowledgments

We would like to acknowledge and thank the children, caregivers and health workers who participated in this study. We would also like to acknowledge TB Alliance for generously donating the XTEMP-R tools.

## Author contributions

**Conceptualization:** Nosivuyile Vanqa, Megan Palmer, Tina Sachs, Rajneesh Taneja, Poonam Pande, Koteswara Rao Inabathina, Anneke C. Hesseling, Anthony J. Garcia-Prats, Graeme Hoddinott.

**Data curation:** Dillon T. Wademan, Willdon J. Filander, Mfundo Mlomzale, Ntokozo Sibisi, Cyril Thwala, Phumlani Memela, Nosivuyile Vanqa, Munira Khan, Graeme Hoddinott.

**Formal analysis:** Dillon T. Wademan, Willdon J. Filander, Mfundo Mlomzale, Anthony J. Garcia-Prats, Graeme Hoddinott.

**Funding acquisition:** Rajneesh Taneja, Anneke C. Hesseling, Anthony J. Garcia-Prats.

**Investigation:** Munira Khan.

**Methodology:** Dillon T. Wademan, Munira Khan, Graeme Hoddinott.

**Project administration:** Megan Palmer, Tina Sachs, Munira Khan.

**Resources:** Rajneesh Taneja, Poonam Pande, Koteswara Rao Inabathina.

**Supervision:** Tina Sachs, Anneke C. Hesseling, Graeme Hoddinott.

**Validation:** Anneke C. Hesseling, Anthony J. Garcia-Prats, Graeme Hoddinott.

**Writing – original draft:** Dillon T. Wademan, Mfundo Mlomzale, Anthony J. Garcia-Prats, Graeme Hoddinott.

**Writing – review & editing:** Dillon T. Wademan, Willdon J. Filander, Mfundo Mlomzale, Ntokozo Sibisi, Nosivuyile Vanqa, Megan Palmer, Munira Khan, Rajneesh Taneja, Poonam Pande, Koteswara Rao Inabathina, Anneke C. Hesseling, Anthony J. Garcia-Prats, Graeme Hoddinott.

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
