## [Decision Letter · Decision Letter 0]

29 Jul 2025

PGPH-D-25-01547

Children, caregivers and health workers’ perceptions and experiences of the XTEMP-R® tool to improve tuberculosis treatment

Dear Dr. Wademan,

Thank you for submitting your manuscript to PLOS Global Public Health. After careful consideration, we feel that it has merit but does not fully meet PLOS Global Public Health’s publication criteria as it currently stands. Therefore, we invite you to submit a revised version of the manuscript that addresses the points raised during the review process.

Please submit your revised manuscript by . If you will need more time than this to complete your revisions, please reply to this message or contact the journal office at globalpubhealth@plos.org. Please include the following items when submitting your revised manuscript:

We look forward to receiving your revised manuscript.

Kind regards,

Kiatichai Faksri, Ph.D

Academic Editor

Journal Requirements:

Additional Editor Comments (if provided):

Subject: Minor Revision Request – PGPH-D-25-01547

Manuscript Title: Children, caregivers and health workers’ perceptions and experiences of the XTEMP-R® tool to improve tuberculosis treatment

Dear Dr. Dillon T. Wademan and authors

Thank you for submitting your manuscript to PLOS Global Public Health. We have now received one reviewer report for your manuscript entitled "Children, caregivers and health workers’ perceptions and experiences of the XTEMP-R® tool to improve tuberculosis treatment" (Manuscript ID: PGPH-D-25-01547).

Based on the reviewer’s feedback, I am recommending that the manuscript undergo minor revision before it can be considered for publication. The reviewer commended the considerable effort evident in your study and found the topic to be of interest and relevance.

However, several clarifications and additions are requested to strengthen the methodological rigor and transparency of your report.

Please address the following key points in your revision:

Justification of Sampling Strategy

Provide a clear rationale for the use of purposeful sampling, particularly for the caregiver participants. Include details on how bias was minimized in the selection process.

Sample Size Determination

Clarify how the sample size for each group/component was determined. If applicable, include theoretical or practical justifications for the chosen sample sizes.

Consent Procedures for Health Care Workers

While the consent process for other participants is described, please also include information on how informed consent was obtained from healthcare workers, for completeness.

Participant Roles in Components 1 and 2

Explain the additional insights obtained from caregivers in component 1 that could not be acquired from participants in component 2. This will help justify the study design and participant allocation.

Use of Software or AI Tools

Indicate whether any software or AI tools were used for data collection, transcription, coding, or analysis. This is in line with transparency expectations from the journal.

Data Availability

In accordance with PLOS’ Data Availability Policy, please ensure that the underlying data supporting your findings are made publicly available or deposited in an appropriate repository. Clarify any restrictions, if applicable.

In preparing your revised submission, please include a detailed response to reviewers, outlining how each comment was addressed.

Once submitted, your revision will be assessed promptly. Please aim to submit your revised manuscript within 30 days of this notice. If additional time is needed, feel free to request an extension.

Thank you again for your submission to PLOS Global Public Health. We look forward to receiving your revised manuscript.

Best regards,

Prof.Kiatichai Faksri

Academic Editor

Reviewers' comments:

Reviewer's Responses to Questions

**Comments to the Author**

1. Does this manuscript meet PLOS Global Public Health’s publication criteria?

Reviewer #1: Partly

Reviewer #2: Yes

2. Has the statistical analysis been performed appropriately and rigorously?

Reviewer #1: I don't know

Reviewer #2: N/A

3. Have the authors made all data underlying the findings in their manuscript fully available (please refer to the Data Availability Statement at the start of the manuscript PDF file)?

Reviewer #1: No

Reviewer #2: Yes

4. Is the manuscript presented in an intelligible fashion and written in standard English?

Reviewer #1: Yes

Reviewer #2: Yes

Reviewer #1: Obviously, a lot of work has gone into this, and I commend the authors. However, there are a few things authors may consider:

1. Purposeful sampling, especially with the caregivers—Provide justification for the sampling technique and how you minimized bias

2. Unclear how authors determined the sample size—include information about sample size determination

3. Good mention of the consent process, but the healthcare workers (HCWs) were not mentioned—include consent process for HCWs for completeness

4. It is unclear what additional information the Caregivers in component 1 provided that could not have been collected as baseline from participants in component 2 - Include justification for the choice of participants in both components

5. Use of AI or any software - Mention if any software or AI was used in data collection, transcription, etc.

Reviewer #2: The overall quality of the manuscript is good. However, a few corrections have to be made as follows.

1. The authors should write in full the following abbreviations on first use:

a. MDR-TB (Introduction line 54)

b. RR/MDR-TB (Introduction line 57)

c. WHO (Introduction line 97)

2. Methods section

a. The authors should include information about the health services delivery in the study site (Setting)

b. The authors should specify the type of interview conducted. For qualitative studies, the typical type of interviews conducted are in-depth interviews and focus group discussions (Data Collection line 121)

**Do you want your identity to be public for this peer review?** For information about this choice, including consent withdrawal, please see our Privacy Policy

Reviewer #1: No

Reviewer #2: **Yes: ** Cyril Kwami Azornu

---

## [Editor Report · Decision Letter 1]

26 Aug 2025

PGPH-D-25-01547R1

Children, caregivers and health workers’ perceptions and experiences of the XTEMP-R tool to improve tuberculosis treatment

Dear Dr. Wademan,

Thank you for submitting your manuscript to PLOS Global Public Health. After careful consideration, we feel that it has merit but does not fully meet PLOS Global Public Health’s publication criteria as it currently stands. Therefore, we invite you to submit a revised version of the manuscript that addresses the points raised during the review process.

We look forward to receiving your revised manuscript.

Kind regards,

Kiatichai Faksri, Ph.D

Academic Editor

Journal Requirements:

Additional Editor Comments (if provided):

Manuscript ID: PGPH-D-25-01547R1

Title: Children, caregivers and health workers’ perceptions and experiences of the XTEMP-R tool to improve tuberculosis treatment

Thank you for your continued efforts in revising the manuscript.

The reviewers appreciate the work you have put into this study. However, one reviewer still feels that the manuscript requires minor revisions before it can be considered further.

Please address the comments below and submit a revised version of your manuscript:

Purposeful sampling – Provide justification for the sampling technique used, particularly with caregivers, and explain how potential bias was minimized.

Sample size determination – Clarify how the sample size was determined and include relevant information in the manuscript.

Consent process for HCWs – While the consent process for other participants is described, please include details for healthcare workers (HCWs) to ensure completeness.

Justification for participant selection – It is unclear what additional information caregivers in Component 1 provided that could not have been obtained as baseline data from participants in Component 2. Please justify the rationale for participant selection in both components.

Use of AI or software – Indicate whether any software tools or AI were used in data collection, transcription, or analysis.

We kindly request that you revise the manuscript accordingly and resubmit it to the journal at your earliest convenience.

Thank you for your attention to these points, and we look forward to receiving your revised submission.

Sincerely,

Kiatichai Faksri

Academic Editor
---

## [Editor Report · Decision Letter 2]

15 Sep 2025

Children, caregivers and health workers’ perceptions and experiences of the XTEMP-R tool to improve tuberculosis treatment

PGPH-D-25-01547R2

Dear Mr Wademan,

We are pleased to inform you that your manuscript 'Children, caregivers and health workers’ perceptions and experiences of the XTEMP-R tool to improve tuberculosis treatment' has been provisionally accepted for publication in PLOS Global Public Health.

Best regards,

Kiatichai Faksri, Ph.D

Academic Editor

Subject: Acceptance of Manuscript PGPH-D-25-01547R2

Dear Dr. Wademan,

I am pleased to inform you that your manuscript entitled:

“Children, caregivers and health workers’ perceptions and experiences of the XTEMP-R tool to improve tuberculosis treatment”

has been accepted for publication in PLOS Global Public Health.

We appreciate the careful attention you and your co-authors gave in addressing the reviewers’ comments and revising the manuscript in line with the journal’s specifications.

Your study makes an important contribution to the field of childhood tuberculosis treatment and offers valuable insights into the acceptability and user experience of the XTEMP-R tool.

Congratulations on this achievement, and thank you for choosing PLOS Global Public Health as the venue for your work. We look forward to seeing your article published.

Sincerely,

Professor Kiatichai Faksri

Academic Editor